# Research on a 3D Point Cloud Map Learning Algorithm Based on Point Normal Constraints

**DOI:** 10.3390/s24196185

**Published:** 2024-09-24

**Authors:** Zhao Fang, Youyu Liu, Lijin Xu, Mahamudul Hasan Shahed, Liping Shi

**Affiliations:** 1School of Mechanical and Automotive Engineering, Anhui Polytechnic University, Wuhu 241000, China; 2220110177@stu.ahpu.edu.cn (Z.F.); shahedhasan201115@gmail.com (M.H.S.); 2Anhui Gongchuang Industrial Robot Innovation Center Co., Ltd., Wuhu 241100, China; xulijin@anhui-at.com (L.X.); shiliping@wh-robot.cn (L.S.)

**Keywords:** point cloud denoising, point normal constraints, Dirichlet energy, Gaussian noise, Laplace noise

## Abstract

Laser point clouds are commonly affected by Gaussian and Laplace noise, resulting in decreased accuracy in subsequent surface reconstruction and visualization processes. However, existing point cloud denoising algorithms often overlook the local consistency and density of the point cloud normal vector. A feature map learning algorithm which integrates point normal constraints, Dirichlet energy, and coupled orthogonality bias terms is proposed. Specifically, the Dirichlet energy is employed to penalize the difference between neighboring normal vectors and combined with a coupled orthogonality bias term to enhance the orthogonality between the normal vectors and the subsurface, thereby enhancing the accuracy and robustness of the learned denoising of the feature maps. Additionally, to mitigate the effect of mixing noise, a point cloud density function is introduced to rapidly capture local feature correlations. In experimental findings on the anchor public dataset, the proposed method reduces the average mean square error (MSE) by 0.005 and 0.054 compared to the MRPCA and NLD algorithms, respectively. Moreover, it improves the average signal-to-noise ratio (SNR) by 0.13 DB and 2.14 DB compared to MRPCA and AWLOP, respectively. The proposed algorithm enhances computational efficiency by 27% compared to the RSLDM method. It not only removes mixed noise but also preserves the local geometric features of the point cloud, further improving computational efficiency.

## 1. Introduction

Point clouds find extensive applications across various domains, including 3D modeling and autonomous driving [1,2,3]. Typically, point cloud data are acquired through two methods [4,5]: 3D scanners and stereo matching algorithms. These methods can introduce significant noise due to matching inaccuracies. The alternative approach employs devices like laser scanners [6], structured light scanners [7], or Lidar [8] for quick real-time point cloud data acquisition. However, point clouds often suffer from noise stemming from both hardware and software sources [9]. This noise not only distorts the underlying structure of the point cloud, impeding surface reconstruction [10] and subsequent tasks, but also increases the presence of irrelevant data and reduces the accuracy of feature extraction, such as point cloud curvature and normal vectors [11].

Noise reduction in point clouds has emerged as a crucial area of focus in processing three-dimensional geometric data. Zeng et al. [12] introduced a method employing graph Laplacian regularization (GLR) for noise reduction. This method extends the low-dimensional manifold model utilized for two-dimensional images to the point cloud surface, treating it as a discrete observation on the manifold. It employs a local region-based graph Laplacian regularizer to approximate the manifold dimension defined in the continuous domain and proposes a novel discrete distance measure to assess the similarity between two equally sized local regions. This measure is then used to construct a noise-resistant graph and facilitate the noise reduction process. Hu et al. [13] proposed a feature map learning paradigm to deduce the underlying graph structure of the point cloud for noise reduction. Unlike the manual adjustment of edge weight parameters, their approach optimizes these parameters strictly through feature measurement learning when the available signal is smooth relative to the graph, thus achieving accurate and rapid noise reduction of the point cloud.

The above algorithms have achieved good results for the additional noise generated on the surface of objects, but they are not suitable for actual mixed noise far away from object noise points. Mixed noise mainly includes Gaussian noise and Laplacian noise. Gaussian noise is usually caused by sensor thermal noise, while Laplacian noise can originate from hardware failures or environmental interference. These types of noise are extremely common in practical applications, and each has different statistical and distribution characteristics, so removing mixed noise points remains challenging.

Therefore, this study proposes a denoising algorithm for Gaussian noise and Laplacian noise to address this challenge. Its model is shown in Figure 1.

This paper presents a novel feature map learning denoising algorithm with point normal constraints to address the aforementioned issues. The key innovations and contributions include:(1)The integration of Dirichlet energy with the coupling orthogonality deviation term to improve the accuracy and robustness of feature map learning denoising;(2)The introduction of a density function to better illustrate the correlation between the local features of the point cloud, preserving the fundamental structure of the denoised point cloud;(3)The design of denoising algorithms for Gaussian noise and Laplace noise, thereby eliminating both types of noise simultaneously.

## 2. Related Work

Currently, noise reduction techniques for point cloud data can be classified into six main categories: sparsity-based methods, non-local methods, techniques employing moving least squares (MLS), approaches based on local optimal projection (LOP), graph-based methods, and those utilizing deep learning. The sparsity method assumes that point cloud data have a sparse representation in a certain transformation domain, thus achieving denoising. It can effectively remove noise and preserve details, but its computational complexity is high and the processing efficiency of large point cloud data is low. Non-local methods perform well in preserving details and edges by identifying and utilizing similar patterns in point clouds for denoising. However, when there is a lot of noise in the point cloud, the accuracy of identifying similar patterns decreases, which affects the denoising effect. The MLS method utilizes moving least squares fitting technology to smooth point cloud data, which is suitable for removing small-scale noise and can smooth the surface while preserving most geometric features. However, its ability to handle complex geometric structures and large-scale noise is limited. The LOP method iteratively optimizes and projects the point cloud onto a more regular surface, effectively removing noise while preserving geometric features. However, the computational cost is high, and in some cases, it may be overly smooth. The graph-based method utilizes graph structure to represent point clouds, and denoising is carried out through graph signal processing techniques, which can effectively handle mixed noise, enhancing denoising accuracy and robustness. However, it requires constructing and processing graph structures, which require high computational resources for very large point cloud data. Deep learning methods learn the features of point cloud denoising by training neural network models. After training on large-scale datasets, they can achieve a balance between accuracy and efficiency, but require a large amount of annotated data for training and require vast computational resources.

### 2.1. Methods Based on Sparsity

This method employs sparse representation aligned with the point’s normal vector, addressing the global issue via sparse regularization and conducting sparse reconstruction in the same direction. It subsequently adjusts the point’s position [14,15] by solving a further global minimization problem, predicated on the local plane assumption. Mattei et al. [16] introduced the mobile robust principal component analysis (MRPCA) method, utilizing a weighted minimization approach to denoise three-dimensional point clouds while preserving sharp features. However, these techniques may yield suboptimal results when local areas with high signal-to-noise ratios produce redundant features, potentially leading to oversmoothing or oversharpening.

### 2.2. Non-Local Methods

Drawing inspiration from the non-local means (NLM) [17] and BM3D [18] image denoising algorithms, the proposed method leverages the self-similarity of point cloud surface patches for noise reduction. Deschaud et al. [19] utilized the polynomial coefficients of the local moving least squares (MLS) surface as neighborhood descriptors to compute point similarity, thereby introducing a non-local denoising (NLD) algorithm. However, this approach depends on the self-similarity among surface blocks in the point cloud and is marked by high computational complexity.

### 2.3. Based on the Moving Least Squares (MLS) Method

Alexa et al. [20] established the point cloud surface through a process of moving least squares projection. This involves projecting any noise points to the least squares surface before smoothing the data. Guennebaud et al. [21] introduced algebraic point set surfaces (APSS), a technique that can inadvertently lead to oversmoothing in the denoising effect.

### 2.4. Methods Based on Local Optimal Projection (LOP)

The local optimum projection (LOP) method aims to minimize noise by projecting a point set onto the natural surface of a point cloud. Huang et al. [22] introduced the weighted local optimum projection (WLOP) algorithm, which enhances the LOP algorithm. While WLOP preserves the point cloud’s features, it tends to be computationally inefficient. Additionally, Huang et al. introduced the anisotropic weighted local optimum projection (AWLOP) algorithm [23]. This algorithm is a modification of WLOP, utilizing an anisotropic weighted function to retain the point cloud’s sharp features. However, it can generate additional features and smooth the results.

### 2.5. Graph-Based Methods

The method of image filtering is based on graph filtering for noise reduction. Dinesh et al. [24] addressed various types of additive noise using two fidelity terms via feature graph Laplacian regularization. Schoenenberger et al. [25] constructed a k-nearest neighbor graph on the input point cloud, subsequently creating a convex optimization problem on the graph, regularized by the point cloud gradient. Zhang et al. [26] encoded the similarity between available pixels as edge weights through feature map learning, and established another corrected similarity between depth images through viewpoint mapping and sparse linear interpolation, achieving posterior denoising of point clouds. Zou et al. [27] extended the low-dimensional manifold model (RSLDM) of two-dimensional images to point cloud surfaces using k-domain partitioning, improved the low dimensional manifold model, and proposed a statistical low-dimensional manifold model. Hu et al. [13] suggested learning feature graphs to optimize edge weights, but they did not consider the constraints of normal vectors, which made it challenging to achieve satisfactory results in complex mixed noise environments.

### 2.6. Methods Based on Deep Learning

Given the constraints of conventional noise reduction algorithms [28] and the success of deep learning [29] in diverse fields in recent years, Qi et al. [30] introduced a pioneering network, PointNet, to directly process raw point cloud data. This addressed the issue that standard deep neural networks necessitate structured inputs, complicating the training process on unordered point clouds. Consequently, it made noise reduction via deep learning feasible. On this foundation, a plethora of outstanding deep learning-based noise reduction techniques have emerged. Among them, PCN [31] is the first network to specialize in noise reduction using deep learning. However, these methods are restricted by the inherent limitations of computer hardware.

## 3. Methods

### 3.1. Normal Point Constraint Feature Map Learning Model

#### 3.1.1. Three-Dimensional Point Cloud and Noise Model

The point cloud is $N={\left\{p_{i}\right\}}_{i=1}^{N}$, where the points in the point cloud are $p_{i}\in R^{3\times N}$. The observation model [27] of the point cloud is shown in Equation (1):(1)N=U+G+L
where N represents the three-dimensional coordinates of N points affected by observational noise, U represents the true three-dimensional coordinates of the point cloud, G represents Gaussian noise, as shown in Equation (2), and L represents Laplacian noise, as shown in Equation (3):(2)$$G\sim N\left(0,\sigma^{2}I\right)$$
where σ represents standard deviation and I represents an identity matrix. Furthermore, U, G, $L\in R^{3\times N}$.
(3)L~N(ϕ,b)
where ϕ represents the position parameter, and $b^{2}=\frac{1}{2}\sigma$ represents the scale parameter. The goal is to restore the observed model N to the ideal model U under the interference of Gaussian noise G and Laplace noise L.

#### 3.1.2. Normal Vector Local Consistency Constraint

To enhance noise removal, the concept of local consistency in point cloud normal vectors is introduced, as depicted in Figure 2. The Dirichlet energy $E_{D}\left(\alpha \right)$ [32] is utilized to penalize the disparities between adjacent normal vectors. This approach ensures the continuity of the point cloud normal vector field, as illustrated in Equation (4):(4)$$E_{D}\left(\alpha \right)=\frac{1}{2}\sum_{i}\sum_{j\epsilon N(i)}{∥\alpha_{i}n_{i}-\alpha_{j}n_{j}∥}^{2}r_{ij}$$
where $n_{i}={\left(n_{i1},n_{i2},n_{i3}\right)}^{T}$ represents the normal vector on point $p_{i}$, $n_{j}={\left(n_{j1},n_{j2},n_{j3}\right)}^{T}$ represents the normal vector on point $p_{j}$, and $p_{i}$ and $p_{j}$ represent two adjacent points on the point cloud block.${\;r}_{ij}=exp(-\frac{∥p_{i-}p_{j}∥}{\theta^{2}})$, θ=Max(qi,qj)ϵξ∥pi−pj∥2.

The coupling orthogonality deviation item $E_{COD}\left(\alpha \right)$ [32] is combined to enhance the orthogonality condition between the normal vector and the bottom surface of point cloud data, as illustrated in Equation (5):(5)$$E_{COD}\left(\alpha \right)=\sum_{i}\sum_{j\epsilon N(i)}r_{ij}{{\left[(\alpha_{i}n_{i}+\alpha_{j}n_{j}\right)}^{T})(\frac{p_{i-}p_{j}}{\alpha_{ij}})]}^{2}$$
where $n_{i}=\alpha_{i}\overset{⃑}{n_{i}}$ and $\alpha_{i}={\left(\alpha_{1},\alpha_{2},\ldots ,\alpha_{m}\right)}^{T}$ are decision vectors.

For point clouds with noisy inputs, the sum of the Dirichlet energy and the coupling orthogonality deviation term $E_{loc}(\alpha )$ is referred to, and the regularizer $L_{1}$ is applied to ensure stability, as shown in Equation (6).
(6)$$E_{loc}\left(\alpha \right)=E_{D}\left(\alpha \right)+E_{COD}\left(\alpha \right)+{∥n_{i}∥}_{1}$$

From Equations (4) and (5), it can be concluded that $E_{D}\left(\alpha \right)$ and $E_{COD}\left(\alpha \right)$ are quadratic functions of $n=({n_{1}}^{T},{n_{2}}^{T},\ldots ,{n_{N}}^{T})$. Therefore, the sum of the Dirichlet energy and the coupling orthogonality deviation term can be represented as shown in Equation (7):(7)$$E_{loc}\left(\alpha \right)=n^{T}Pn$$
where P represents a sparse matrix. For denoising a point cloud with noisy input, the constraint of local consistency in the normal vector ensures that its local features are preserved after denoising.

#### 3.1.3. Probability Density Function

Each point in the point cloud, labeled as point $c_{i}$, is used to construct a point cloud block by identifying its k nearest neighbors. To meet the self-similarity requirement of the point cloud block, it must be translated and aligned, ensuring that the central point of each block is situated at the origin. The mathematical representation of the point cloud block [13] is provided in Equation (8):(8)V=(TN−C)
where $T\in {\left(0,1\right)}^{\left(k+1\right)M\times N}$ is the selection matrix, $C={\left\{c_{i}\right\}}_{i=1}^{M}$, and points are selected from the point cloud N to form M point cloud blocks, each of which contains k+1 points.

To accommodate the complexities of object point cloud denoising, the density function $\zeta =f(c_{i})$ is introduced, and Equation (9) is derived as follows:(9)V=ζ(TN−C)

### 3.2. Maximum A Posteriori (MAP) Point Cloud Denoising

According to reference [13], the problem of estimating point cloud blocks, as shown in Equation (10), is derived as follows:(10)$$\upsilon_{map}\left(V\right)=\underset{V}{\mathrm{argmax}}f(\upsilon \left|V\right)g(V)$$
where $f\left(\upsilon |V\right)$ represents the likelihood function and $g\left(V\right)$ is the prior probability distribution of V.

Point cloud blocks are derived from observations, making $f\left(\upsilon |V\right)$ and f(N|U) equivalent. Consequently, the likelihood function $f\left(\upsilon |V\right)$ [13], formulated based on the noise model, is employed in Equations (7)–(9), as depicted in Equation (11):(11)$$f\left(\upsilon |V\right)=f(N|U)={exp\{-∥N-U∥}_{F}^{2}\}$$

According to reference [13], g(V) is obtained, as shown in Equation (12):(12)$$g\left(V\right)=exp\{-\beta tr(V^{T}L\left(M\right)V)\}$$
where $\beta =\frac{1}{{2\sigma }^{2}}$ represents parameters related to noise intensity, and M [13] is the Mahalanobis distance matrix between two points.

An overall Equation (13) for point cloud denoising can be obtained by combining Equations (9)–(12):(13)$$\underset{U,M}{\mathrm{argmax}}{exp\{-∥U-N∥}_{F}^{2}-{∥N_{i}-n_{i}∥}_{F}^{2}-\beta tr({\left(\zeta (TN-C\right)}^{T}L\left(M\right)\zeta \left(TN-C\right)$$

Taking the logarithm of Equation (13) and multiplying by −1:(14)$$\underset{U,M}{\mathrm{argmin}}{∥U-N∥}_{F}^{2}+{∥N_{i}-n_{i}∥}_{F}^{2}+\beta tr\left({\left(\zeta (TN-C\right)}^{T}L\left(M\right)\zeta \left(TN-C\right)\right)$$

For any point $p_{i}$ in point cloud N, let $S_{i}$ represent the average distance to k points in its neighborhood. The distance threshold [31] is obtained as shown in Equation (15).
(15)$$d_{\mathrm{threshold}\;}=\frac{\sum_{i=1}^{n}\;S_{i}}{N}\pm std\frac{\sum_{i=1}^{n}\;{\left(S_{i}-\frac{\sum_{i=1}^{n}\;S_{i}}{N}\right)}^{2}}{N}$$

The threshold of standard deviation multiples is represented by the number std. The distance between point $p_{i}$ and its neighborhood, which is outside of $(\frac{\sum_{i=1}^{n}\;S_{i}}{N}-std\cdot \sum_{i=1}^{n}\;{\left(S_{i}-\frac{\sum_{i=1}^{n}\;S_{i}}{N}\right)}^{2}/N$ and $\frac{\sum_{i=1}^{n}\;S_{i}}{N}+std\cdot \sum_{i=1}^{n}\;{\left(S_{i}-\frac{\sum_{i=1}^{n}\;S_{i}}{N}\right)}^{2}/N)$, is considered a Laplace noise point. Therefore, the location matrix 3 of the Laplace noise points is determined by statistical methods.

### 3.3. Point Cloud Noise Removal Solution

The problem in Equation (14) is solved by optimizing the point cloud U, the point cloud normal vector $N_{i}$, and the Mahalanobis distance matrix M between the two points.

Optimizing point cloud U and point cloud normal vector $N_{i}$: Initialize the Mahalanobis distance matrix M with the identity matrix I, and fix the combination Laplacian matrix L in Equation (14). Taking the derivative of the point cloud U and the normal vector $N_{i}$ of the point cloud in the X direction,
(16)$$\left(\beta T^{T}LT+I\right)U_{x}+\left(\beta T^{T}LT+I\right)N_{ix}=N_{x}+n_{ix}+\beta T^{T}LC_{x}$$

Taking the derivative of the point cloud U and the normal vector $N_{i}$ of the point cloud in the Y direction,
(17)$$\left(\beta T^{T}LT+I\right)U_{y}+\left(\beta T^{T}LT+I\right)N_{iy}=N_{y}+n_{iy}+\beta T^{T}LC_{y}$$

Taking the derivative of the point cloud U and the normal vector $N_{i}$ of the point cloud in the Z direction,
(18)$$\left(\beta T^{T}LT+I\right)U_{z}+\left(\beta T^{T}LT+I\right)N_{iz}=N_{z}+n_{iz}+\beta T^{T}LC_{z}$$

Equations (15)–(17) form a system of linear equation. These are solved iteratively using the conjugate gradient algorithm. Subsequently, the derived U solution is used to update M in the following iterations.

## 4. Algorithm Design

The solution process based on the point cloud denoising model, using feature maps with point normal constraints to learn denoising as shown in Algorithm 1.
**Algorithm 1**: Denoising for 3D point cloud based on PNCFGLInput: N, n, V, C, ζ.Output: Denoised cloud U.1: Initializing U with N;2: for iter =1, 2, … r do;3: Sampling s points from U as a block center;4: Find the k nearest neighbors of the center of each block to form a surface block;5: Optimizing objective function: argminO,M⁡∥U−N∥F2+∥Ni−ni∥F2+βtr((ζ(TN−C)TLMζ(TN−C)), subject to: N=U+G+L;6: L←dthreshold =∑i=1n SiN±std⋅∑i=1n Si−∑i=1n SiN2N;7:ζ←f(ci)8: U converges;9: **end for**

## 5. Performance Evaluation

Performance evaluation of denoising algorithms can be quantitatively assessed through signal-to-noise ratio (SNR) [12] and mean squared error (MSE) [26].

The signal-to-noise ratio (SNR) is defined as shown in Equation (19):(19)$$SNR\left(\varpi_{1},\varpi_{2}\right)10lg\frac{\frac{2}{N_{2}}\sum_{u_{i}\in }\varpi_{2}{∥u_{j}∥}_{2}^{2}}{\frac{1}{N_{1}}\sum_{u_{i}\in }\varpi_{1}\underset{u_{j}\in \varpi_{2}}{\min}{∥u_{i}-u_{j}∥}_{2}^{2}+\frac{1}{N_{2}}\sum_{u_{i}\in }\varpi_{2}\underset{u_{j}\in \varpi_{1}}{\min}{∥u_{i}-u_{j}∥}_{2}^{2}}$$

The mean squared error (MSE) is defined as shown in Equation (20):(20)$$MSE\left(\varpi_{1},\varpi_{2}\right)\frac{1}{2}(\frac{1}{N_{1}}\sum_{u_{i}\in }\varpi_{1}\underset{u_{j}\in \varpi_{2}}{\min}{∥u_{i}-u_{j}∥}_{2}^{2}+\frac{1}{N_{2}}\sum_{u_{i}\in }\varpi_{2}\underset{u_{j}\in \varpi_{1}}{\min}{∥u_{i}-u_{j}∥}_{2}^{2})$$
where $\varpi_{1}={\left\{u_{i}\right\}}_{i=1}^{N_{1}},u_{i}\in R^{3},\varpi_{2}={\left\{u_{j}\right\}}_{i=1}^{N_{2}},u_{j}\in R^{3}$.

The larger the SNR, the lower the distortion of point cloud denoising and the better the denoising effect. The smaller the MSE, the lower the deviation between the ground truth value and the denoised point cloud and the better the denoising effect.

## 6. Results and Discussion

### 6.1. Comparison and Analysis of Noise Reduction Performance

To verify the effectiveness of the proposed algorithm, the common point cloud model designed by Anchor and Gargoyle, which has high-density features, was used for denoising research. Therefore, density partitioning in the proposed algorithm can be effectively performed. One may then compare the performance with existing algorithms under different noise intensity conditions.

The ground truth point cloud model for the anchor comprises 55,799 data points, and for the gargoyle, it contains 58,611 data points, as depicted in Figure 3. Gaussian and Laplace noise, with means of 0 and standard deviations of 0.02, 0.03, 0.04, and 0.05, are introduced to the ground truth point cloud model. The cases where σ = 0.02 and σ = 0.04 are illustrated in Figure 4. To facilitate the illustration of the effects post-addition and denoising, color information is incorporated into the generated model, with the color bar denoting depth information. The feature map learning algorithm, integrating point normal constraints as proposed in this paper, is employed for the denoising of 3D point cloud models across various noise intensities, the outcomes are presented in Figure 5 and Figure 6. From the denoising results, it is evident that the algorithm proposed herein effectively eliminates Gaussian and Laplace noise from the public point cloud while appropriately preserving their original geometric features. Consequently, it constitutes an efficacious point cloud denoising algorithm.

The noisy point cloud models at varying intensities are presented in Figure 4, denoised individually by APSS, NLD and MRPCA, with corresponding results depicted in Figure 7, Figure 8, Figure 9, Figure 10, Figure 11 and Figure 12. In Figure 7 and Figure 8, uneven color distribution and blurred planar features are observed on the surface of the denoised point cloud model by the APSS algorithm. Additionally, the increased presence of outlier noise is noted. Comparatively, Figure 9 and Figure 10, as well as Figure 11 and Figure 12, display relatively uniform color distribution on the surfaces. However, some noise persists away from the main body. The analysis of the results in Figure 7, Figure 8, Figure 9, Figure 10, Figure 11 and Figure 12 indicates inadequate elimination of Laplace noise. Subsequently, the PNCFGL algorithm was used for noise removal, and the results showed that it significantly reduced abnormal noise and improved the accuracy of local features.

As depicted in Table 1, under the MSE metrics, an anchor-based point cloud model after denoising at four noise levels with different values of σ (0.02–0.05), at σ = 0.05, the anchor is 0.267 for APSS, 0331 for NLD, 0.315 for AWLOP, and 0.253 for MRPCA and 0.245 for the algorithm in this paper, which is a reduction of 0.022 compared to APSS, 0.086 compared to NLD, 0.070 compared to AWLOP, and 0.008 compared to MRPCA. After denoising, the average MSE of APSS, AWLOP, NLD, and MRPCA are 0.242, 0.280, 0.281, and 0.232, respectively, and the average MSE of the algorithm in this paper is 0.227, which is reduced by 0.015 compared to APSS, 0.054 compared to NLD, 0.053 compared to AWLOP, and 0.005 compared to MRPCA. Quantitative analysis underscores the superior denoising efficacy of this paper’s algorithm, characterized by lower deviation between the ground truth value and the denoised point cloud.

As depicted in Table 2, under the MSE metrics, an anchor-based point cloud model after denoising at four noise levels with different values of σ (0.02–0.05), at σ = 0.05, the anchor is 47.09 DB for APSS, 44.94 DB for NLD, 45.44 DB for AWLOP, 47.64 DB for MRPCA and 47.81 DB for the algorithm in this paper, which is an improvement of 0.72 DB over APSS, 2.87 DB over NLD, 2.37 DB over AWLOP, and 0.17 DB over MRPCA. After denoising, the average SNR of APSS, NLD, AWLOP, and MRPCA are 48.14 DB, 46.66 DB, 46.54 DB, and 48.55 DB, respectively. The average SNR of the algorithm in this paper is 48.68 DB, which is 0.54 DB better than APSS, 2.14 DB better than AWLOP, 2.02 DB better than NLD, and 0.13 DB better than MRPCA. It is quantitatively analyzed to be better than the first three algorithms, with the lowest distortion in the denoising process.

In terms of the comparison of noise point clouds under different intensities with the latest graph learning algorithm RSLDM, according to Table 1 and Table 2, the average MSE of RSLDM is 0.230, while the algorithm in this paper is 0.227, which is an improvement of 0.003. The average SNR of RSLDM is 48.62 DB, while the algorithm proposed in this paper is 48.68 DB, which is an improvement of 0.06 DB. However, this algorithm has improved computational efficiency by 27% compared to RSLDM due to RSLDM dividing point clouds into blocks based on the k-neighborhood, while our algorithm divides point clouds into blocks based on density. Density partitioning utilizes the distribution density of points in the point cloud to determine the partitioning boundary, without the need to calculate the k-neighborhood of each point, reducing computational complexity and time consumption. Furthermore, it can determine the partition boundary based on the actual density distribution of point cloud data, which performs better when dealing with point clouds with uneven distribution or large-scale density changes. The impact of density partitioning on noise is relatively small. It determines the boundary through the overall density distribution, rather than relying solely on the local neighborhood of each point.

The algorithm in this article combines point cloud density information and the consistency of normal vector direction to simultaneously remove Gaussian noise and Laplacian noise. Density information can help identify noise points in sparse regions, while normal vector information can help identify noise points with inconsistent normal vectors. Combining these two types of information can effectively combat different types of noise and improve the accuracy and reliability of point cloud data.

### 6.2. Noise Reduction Applications of 3D Point Clouds

To further validate the proposed PNCFGL model, a Lidar scanner was used to capture point clouds of four objects, as shown in Figure 13. The four objects were pipe joints, pipe rings, rear brackets, and front brackets, as shown in Figure 14. After denoising, the MSE and SNR results are shown in Table 3 and Table 4. The environment of the denoising equipment used was as follows: the CPU model was AMD Ryzen7 3800X (Santa Clara, CA, USA), with a basic frequency of 3.90 GHz; the graphics card model was RTX3090 (Nvidia Corporation, Santa Clara, CA, USA); and the memory was 32 GB DDR4.

Gaussian noise and Laplacian noise, each with mean values of 0 and standard deviations of 0.02, 0.03, 0.04, and 0.05, were introduced into the scanned point cloud. The denoising performance of PNCFGL was then evaluated using mean squared error (MSE) and signal-to-noise ratio (SNR), as illustrated in Figure 15. Post-PNCFGL denoising, the outlier noise associated with the four objects witnessed a notable reduction, while preserving the local feature structure effectively.

As depicted in Table 3, under the MSE metrics, at σ = 0.02, the object a is 0.052 for APSS, 0.075 for NLD, 0.081 for AWLOP, and 0.047 for MRPCA and 0.043 for the algorithm in this paper, which is a reduction of 0.009 compared to APSS, 0.032 compared to NLD, 0.038 compared to AWLOP, and 0.004 compared to MRPCA. At σ = 0.03, the object b is 0.075 for APSS, 0.104 for NLD, 0.096 for AWLOP, and 0.068 for MRPCA and 0.063 for the algorithm in this paper, which is a reduction of 0.012 compared to APSS, 0.041 compared to NLD, 0.033 compared to AWLOP, and 0.005 compared to MRPCA. At σ = 0.04, the object c is 0.090 for APSS, 0.130 for NLD, 0.142 for AWLOP, and 0.078 for MRPCA and 0.073 for the algorithm in this paper, which is a reduction of 0.012 compared to APSS, 0.057 compared to NLD, 0.069 compared to AWLOP and 0.005 compared to MRPCA. At σ = 0.05, the object d is 0.108 for APSS, 0.170 for NLD, 0.156 for AWLOP, and 0.094 for MRPCA and 0.084 for the algorithm in this paper, which is a reduction of 0.024 compared to APSS, 0.086 compared to NLD, 0.072 compared to AWLOP, and 0.010 compared to MRPCA. Quantitative analysis reveals that the algorithm in this study demonstrates reduced point cloud deviation and enhanced performance post-noise removal.

As depicted in Table 4, under the SNR metrics, at σ = 0.02, the object a is 83.75 DB for APSS, 82.65 DB for NLD, 82.45 DB for AWLOP, and 84.02 DB for MRPCA and 84.18 DB for the algorithm in this paper, which is an improvement of 0.43 DB over APSS, 1.53 DB over NLD, 1.73 DB over AWLOP, and 0.16 DB over MRPCA. At σ = 0.03, the object b is 76.63 DB for APSS, 76.55 DB for NLD, 76.16 DB for AWLOP, 78.06 DB for MRPCA and 78.19 DB for the algorithm in this paper, which is an improvement of 1.56 DB over APSS, 1.64 DB over NLD, 2.03 DB over AWLOP, and 0.13 DB over MRPCA. At σ = 0.04, the object c is 78.66 DB for APSS, 77.09 DB for NLD, 76.86 DB for AWLOP, 79.18 DB for MRPCA and 79.24 DB for the algorithm in this paper, which is an improvement of 0.58 DB over APSS, 2.15 DB over NLD, 2.38 DB over AWLOP, and 0.06 DB over MRPCA. At σ = 0.05, the object d is 76.52 DB for APSS, 74.45 DB for NLD, 74.92 DB for AWLOP, 77.12 DB for MRPCA and 77.28 DB for the algorithm in this paper, which is an improvement of 0.76 DB over APSS, 2.83 DB over NLD, 2.36 DB over AWLOP, and 0.16 DB over MRPCA. Quantitative analysis indicates that the denoising procedure implemented by the algorithm proposed in this paper exhibits diminished distortions and superior denoising efficacy.

## 7. Conclusions

In this study, a novel point normal constrained feature map learning approach is proposed for effectively mitigating mixed noise within point clouds. Specifically, the reinforcement of normal vectors with subsurface orthogonality conditions is achieved through the incorporation of Dirichlet energy and coupled orthogonality bias terms, aimed at penalizing differences among neighboring normal vectors. These imposed constraints significantly enhance the accuracy and robustness of feature map learning, thus facilitating the denoising of point clouds exhibiting prominent geometric features. Denoising efficacy is evaluated through quantitative metrics, including signal-to-noise ratio (SNR) and mean squared error (MSE), applied to the denoising of the anchor model across four distinct noise levels. Compared with APSS, AWLOP, NLD, MRPCA, and RSLDM, the proposed algorithm has an average SNR improvement of 2.14 DB and an average MSE decrease of 0.054, while achieving similar computational efficiency. While the denoising performance and RSLDM have slightly improved, the computational efficiency has increased by 27%. This algorithm achieves a balance between denoising performance and computational efficiency. Therefore, this algorithm not only effectively removes mixed noise, but also preserves and improves computational efficiency, providing a feasible solution for point cloud data processing. However, due to the need for point cloud density partitioning during its denoising process, it is currently only applicable to high-density point cloud features. Future work will focus on extending this method to low-density application scenarios, further optimizing the algorithm to meet different practical application needs, improving the real-time performance and stability of the algorithm and exploring more constraints and regularization terms to further improve the effectiveness and accuracy of point cloud denoising.

## Figures and Tables

**Figure 1 sensors-24-06185-f001:**
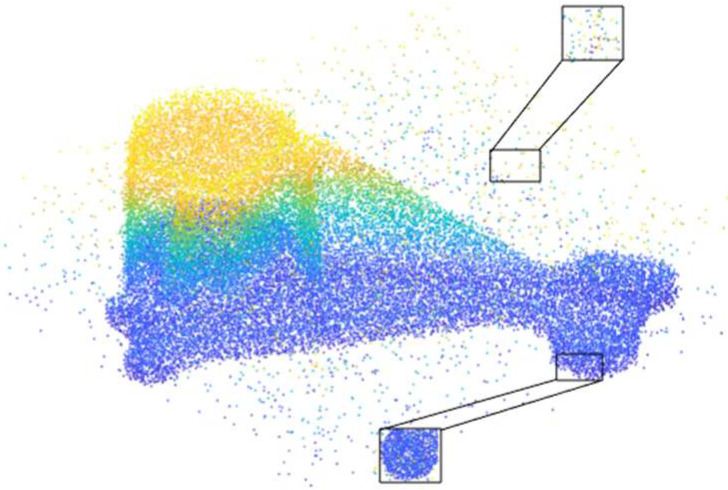
A point cloud model with Gaussian noise and Laplacian noise.

**Figure 2 sensors-24-06185-f002:**
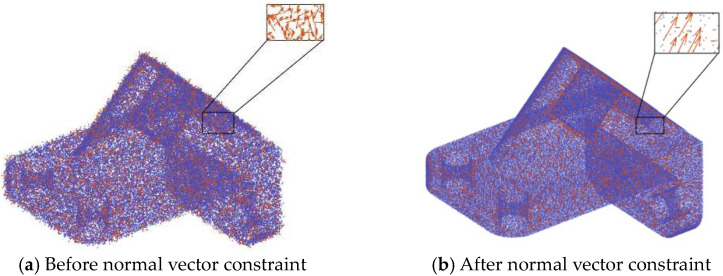
Local consistency constraint of point cloud normal vectors.

**Figure 3 sensors-24-06185-f003:**
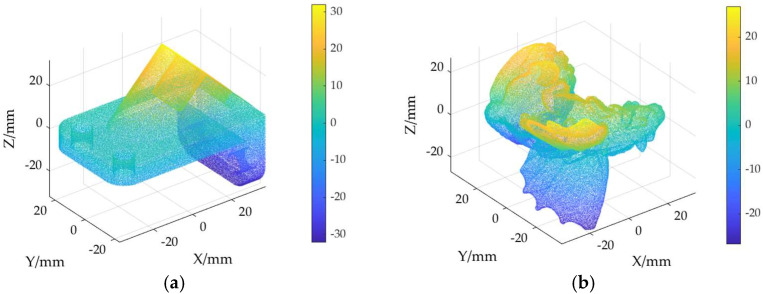
Point cloud model: (**a**) Anchor ground truth; (**b**) Gargoyle ground truth.

**Figure 4 sensors-24-06185-f004:**
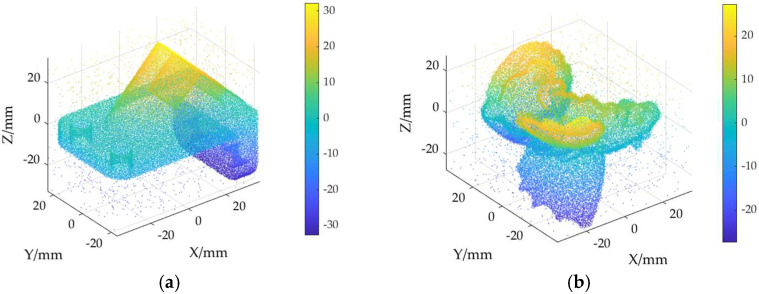
Point cloud model at different noise strengths: (**a**) Noise (σ = 0.02); (**b**) Noise (*σ* = 0.04).

**Figure 5 sensors-24-06185-f005:**
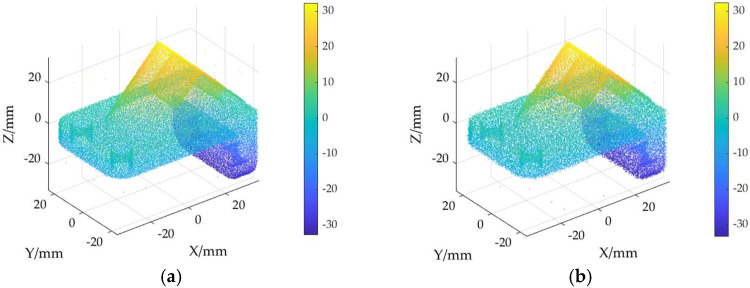
Noise reduction effects of PNCFGL at different strengths in Anchor model: (**a**) PNCFGL (σ = 0.02); (**b**) PNCFGL (*σ* = 0.04).

**Figure 6 sensors-24-06185-f006:**
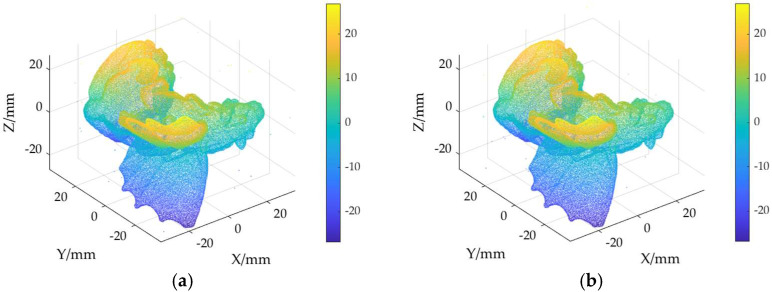
Noise reduction effect of PNCFGL at different strengths in Gargoyle model: (**a**) PNCFGL (σ = 0.02); (**b**) PNCFGL (*σ* = 0.04).

**Figure 7 sensors-24-06185-f007:**
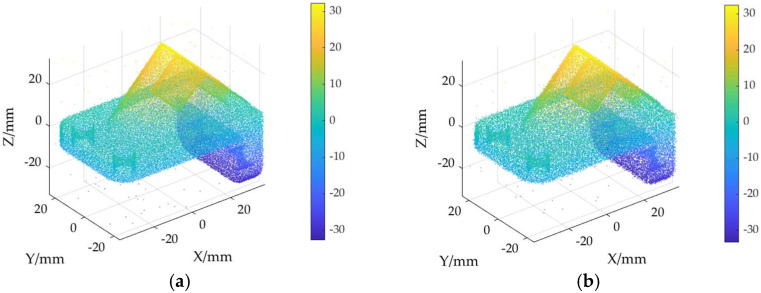
Noise reduction effect of APSS at different noise strengths in Anchor model: (**a**) APSS (σ = 0.02); (**b**) APSS (*σ* = 0.04).

**Figure 8 sensors-24-06185-f008:**
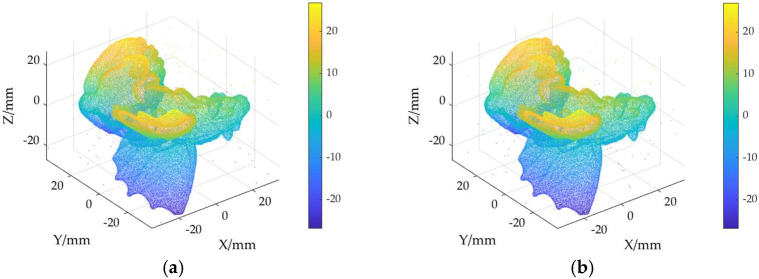
Noise reduction effect of APSS at different noise intensities in Gargoyle model: (**a**) APSS (σ = 0.02); (**b**) APSS (*σ* = 0.04).

**Figure 9 sensors-24-06185-f009:**
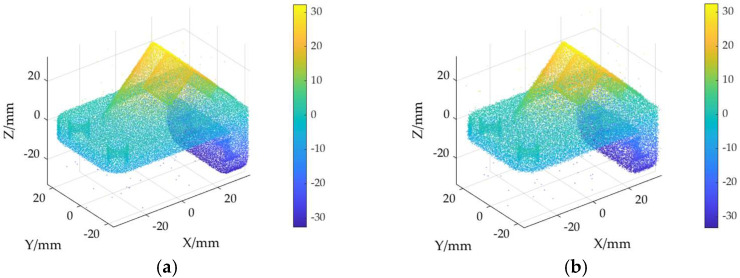
Noise reduction effects of NLD at different strengths in Anchor model: (**a**) NLD (σ = 0.02); (**b**) NLD (*σ* = 0.04).

**Figure 10 sensors-24-06185-f010:**
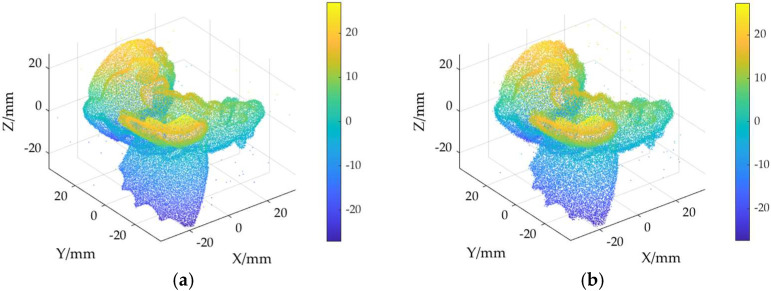
Noise reduction effects of NLD at different strengths in Gargoyle model: (**a**) NLD (σ = 0.02); (**b**) nld (*σ* = 0.04).

**Figure 11 sensors-24-06185-f011:**
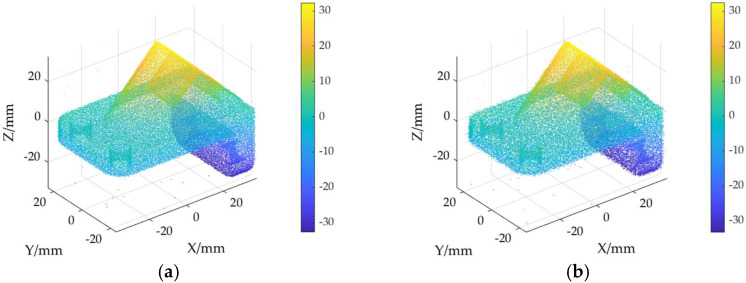
Noise removal effect of MRPCA at different strengths in Anchor model: (**a**) MRPCA (σ = 0.02); (**b**) MRPCA (*σ* = 0.04).

**Figure 12 sensors-24-06185-f012:**
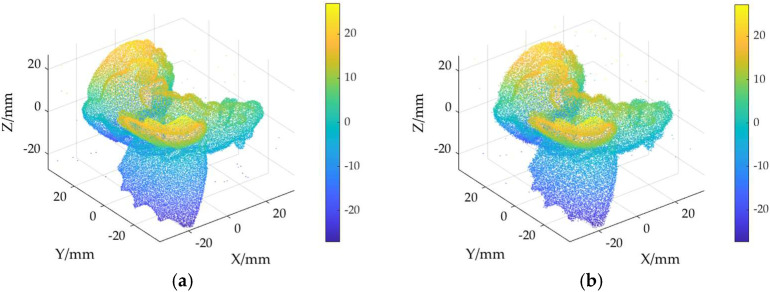
De-noising effects of MRPCA at different strengths in Gargoyle model: (**a**) MRPCA (σ = 0.02); (**b**) MRPCA (*σ* = 0.04).

**Figure 13 sensors-24-06185-f013:**
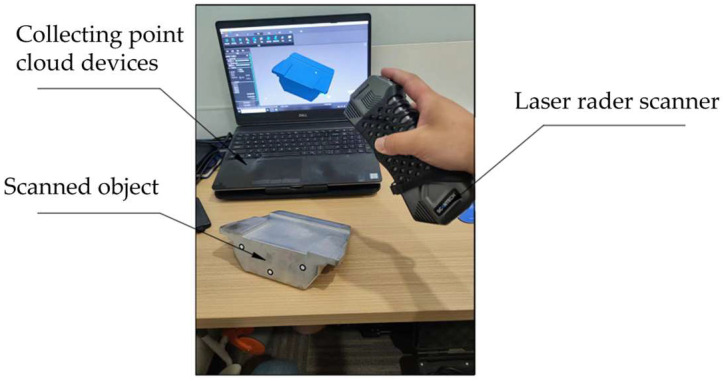
Laser radar scanner.

**Figure 14 sensors-24-06185-f014:**
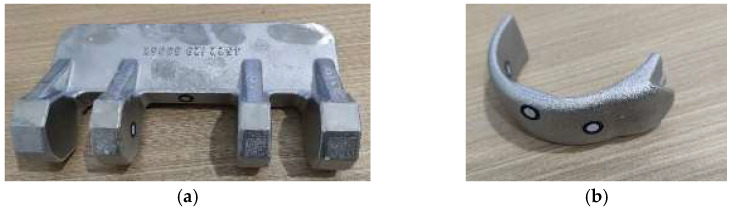
Scanning objects: (**a**) pipe fitting; (**b**) pipe circle; (**c**) rear support; (**d**) front support.

**Figure 15 sensors-24-06185-f015:**
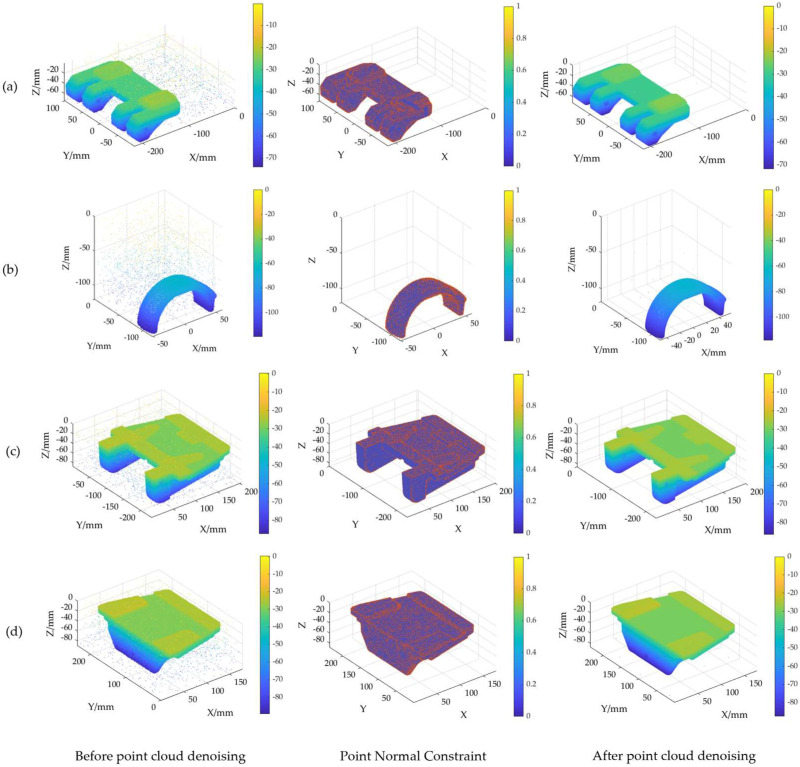
Comparison of objects (**a**–**d**) before and after PNCFGL noise reduction.

**Table 1 sensors-24-06185-t001:** MSE of five different denoising algorithms for the anchor model.

Noise Levels	σ = 0.02	σ = 0.03	σ = 0.04	σ = 0.05	Average	Time
Noisy	0.259	0.322	0.372	0.417	0.343	754 s
APSS	0.208	0.239	0.254	0.267	0.242	891 s
AWLOP	0.237	0.259	0.306	0.315	0.280	780 s
NLD	0.231	0.265	0.297	0.331	0.281	718 s
MRPCA	0.202	0.230	0.242	0.253	0.232	868 s
RSLDM	0.201	0.228	0.240	0.249	0.230	1073 s
Ours	0.199	0.226	0.237	0.245	0.227	782 s

**Table 2 sensors-24-06185-t002:** SNR of five different denoising algorithms for the anchor model.

Noise Level	σ = 0.02	σ = 0.03	σ = 0.04	σ = 0.05	Average	Time
Noisy	47.41	45.25	43.18	42.65	44.62	754 s
APSS	49.61	48.24	47.60	47.09	48.14	891 s
NLD	48.53	47.16	46.02	44.94	46.66	780 s
AWLOP	48.31	46.69	45.74	45.44	46.54	718 s
MRPCA	49.88	48.60	48.09	47.64	48.55	868 s
RSLDM	49.97	48.65	48.12	47.76	48.62	1073 s
Ours	50.04	48.70	48.15	47.81	48.68	782 s

**Table 3 sensors-24-06185-t003:** Comparison of MSE for objects a–d under different algorithms.

Method	Noisy	APSS	NLD	AWLOP	MRPCA	RSLDM	Ours
σ = 0.02							
a	0.103	0.052	0.075	0.081	0.047	0.046	0.043
b	0.116	0.065	0.083	0.092	0.057	0.057	0.054
c	0.119	0.061	0.080	0.089	0.056	0.054	0.052
d	0.113	0.063	0.082	0.091	0.057	0.056	0.053
Average	0.112	0.060	0.080	0.088	0.054	0.053	0.505
σ = 0.03							
a	0.160	0.077	0.102	0.092	0.065	0.064	0.062
b	0.159	0.075	0.104	0.096	0.068	0.063	0.063
c	0.161	0.079	0.103	0.098	0.069	0.066	0.064
d	0.160	0.076	0.102	0.091	0.065	0.063	0.062
Average	0.160	0.076	0.103	0.094	0.067	0.064	0.063
σ = 0.04							
a	0.207	0.089	0.132	0.141	0.077	0.075	0.072
b	0.195	0.078	0.121	0.130	0.064	0.064	0.062
c	0.204	0.090	0.130	0.142	0.078	0.077	0.073
d	0.203	0.088	0.127	0.140	0.075	0.073	0.071
Average	0.202	0.086	0.128	0.138	0.074	0.072	0.070
σ = 0.05							
a	0.253	0.103	0.167	0.151	0.089	0.087	0.081
b	0.242	0.096	0.158	0.141	0.082	0.076	0.073
c	0.254	0.102	0.168	0.153	0.091	0.086	0.082
d	0.257	0.108	0.170	0.156	0.094	0.088	0.084
Average	0.252	0.102	0.166	0.150	0.089	0.084	0.080

**Table 4 sensors-24-06185-t004:** Comparison of SNR for objects a–d under different algorithms.

Method	Noisy	APSS	NLD	AWLOP	MRPCA	RSLDM	Ours
σ = 0.02							
a	81.55	83.75	82.65	82.45	84.02	84.10	84.18
b	76.32	78.30	77.32	76.69	78.69	78.75	78.83
c	79.50	81.63	80.25	80.33	82.01	82.10	82.16
d	78.68	80.88	79.90	79.58	81.12	81.25	81.31
Average	79.01	81.14	80.03	79.76	81.46	81.55	81.58
σ = 0.03							
a	77.08	80.07	78.99	78.50	80.43	80.44	80.53
b	74.64	77.63	76.55	76.16	78.06	78.12	78.19
c	77.34	80.23	79.09	78.68	80.56	80.61	80.69
d	76.56	79.39	78.30	77.89	79.78	79.87	79.91
Average	76.40	79.33	78.23	77.81	79.71	79.76	79.83
σ = 0.04							
a	74.02	78.56	76.90	76.62	78.99	79.01	79.05
b	71.66	76.08	74.55	74.25	76.54	76.54	76.63
c	74.17	78.66	77.09	76.86	79.18	79.20	79.24
d	73.52	77.96	76.42	76.32	78.43	78.47	78.53
Average	73.34	77.82	76.24	76.01	78.29	78.28	78.36
σ = 0.05							
a	72.52	76.96	74.81	75.31	77.51	77.56	77.68
b	70.06	74.50	72.45	72.79	75.06	75.10	75.20
c	72.65	77.19	75.12	75.54	77.86	77.87	77.91
d	72.15	76.52	74.45	74.92	77.12	77.18	77.28
Average	71.85	76.29	74.21	74.64	76.89	76.92	77.02

## Data Availability

The data presented in this study are available on request from the corresponding author.

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
