# Peer review of "Research on a 3D Point Cloud Map Learning Algorithm Based on Point Normal Constraints"

_sensors, 2024, doi:10.3390/s24196185_

Round 1
Reviewer 1 Report
Comments and Suggestions for Authors
(1) The paper presents some useful work, but there are many shortcomings. It is hoped that the authors will be able to improve the quality of their paper before it is considered for submission. My general comments can be summarised as follows:
(2) Line 47: The reference is incorrectly cited. For example, Hu et al [13]. proposed a feature map... , the correct way to write it should be Hu et al. [13] proposed a feature map..... There are similar errors elsewhere in the paper.
(3) The Non-Local Denoising (NLD) algorithm in line 87 without citation. The Advanced Weighted Local Optimum Projection (AWLOP) algorithm in line 101 is also not cited. These two algorithms are compared experimentally with the proposed method and are important enough to be cited.
(4) The proposed method is experimentally compared with the methods of APSS [2007], MRPCA [2017], NLD [2018 or possibly earlier] and AWLOP [2009], but these methods are not the most recent point cloud denoising methods. Also, the method in this paper is based on a graph feature learning algorithm, why not compare it with other graph feature learning algorithms. How can you show the novelty of your method without comparing it with the latest methods in the last two years?
(5) Lines 136-137: Are the distributions of Gaussian and Laplace noise the same? If not, are equations (2) and (3) correct?
(6) Is 0.02 in Figure 4 the intensity of Gaussian noise or Laplace noise? Or is the intensity 0.02 for both types of noise?
(7) The proposed method removes both types of noise at the same time. How well does it remove each type of noise? Does the algorithm remove both types of noise in exactly the same way? There are no experiments and discussion in the paper.
(8) The proposed algorithm depends on several parameters that need to be discussed. It is therefore unclear how the choice of such parameters would affect the overall performance of the proposed framework.
(9) Lines 199 and 203: The variable P should be in italics.
(10) The readability of the paper (especially the results and discussion) could be improved.
Comments on the Quality of English LanguageMinor editing of English language required.
Reviewer 2 Report
Comments and Suggestions for Authors
The denoising of point clouds is a significant issue during the model reverse modeling and visualization. This paper proposed a novel denoising algorithm by combining point-normal constraints, Dirichlet energy, and coupled orthogonality bias terms, to improve the average signal-to-noise ratio. The result shows that the proposed method can effectively remove hybrid noise as well as preserve the local geometric features.
The work implemented in this paper is interesting and valuable for the freeform surface automatic inspection, however, several issues need to be stated before publication. The content needs careful rearrangement to highlight the most significant scientific contributions of this work. And the paper quality could be further improved with the following considerations.
1. In the introduction, the author mentioned that the removal of the complex mixed noise points remains a significant challenge. What the complex mixed noise is is not introduced. Subsequently, a novel denoising algorithm for Gaussian noise and Laplace noise is proposed. And why the Gaussian noise and Laplace noise are chosen as the target is not explained. A detailed and comprehensive investigation of the state of the art can help highlight the most significant contributions of this work.
2. In the Related work, several noise reduction techniques for point cloud data are introduced. A table for the comparison of the noise reduction is suggested, which can clearly demonstrate the advantages and disadvantages of the accuracy, efficiency, robustness, etc. for readers to better follow.
3. To improve the average signal-to-noise ratio, the normal vector local consistency constraint is taken into consideration. Since the original point cloud contains Gaussian noise 𝑮 and Laplace noise 𝑳, how do the authors obtain the accurate normal vector based on the inaccurate point cloud?
4. In the simulation and experimental, the result comparison of different denoising methods is conducted. However, only the signal-to-noise ratio and mean squared error are listed. Besides the accuracy, the time consumption is also significant. No information on calculation efficiency is provided, which can help further verify the feasibility and validity of the proposed algorithm.
5. The authors mention that the proposed algorithm can effectively remove hybrid noise as well as preserve the local geometric features. However, there is no detailed quantifiable information on the reversing of local geometric features. Moreover, the normal vector of the original is also not provided.
6. In this paper, a novel denoising algorithm by combining point-normal constraints, Dirichlet energy, and coupled orthogonality bias terms, to improve the average signal-to-noise ratio. However, the title stressed Point Cloud Feature Map Learning, which is suggested to further consider. Moreover, there is little about the construction of the Feature Map Learning model.
Reviewer 3 Report
Comments and Suggestions for Authors
The method proposed in this paper is innovative and practical. The proposed method can effectively eliminate Gaussian and Laplacian noise; however, this article has more formatting and typographical problems that should be corrected:
Lines 32 and 483: This paper cites some arXiv papers. These arXiv papers are not peer-reviewed. So you should replace the arXiv papers (if any) (unless they are very related to the research area of the article) with related articles from high-impact factor journals, e.g. the fifth article in Ref.
Lines 34, 377 and 488: should be case sensitive for Lidar (Light Detection and Ranging).
Lines 374 and 375 (similar problems occur in lines 264 and 265): The formatting of the table captions in Table1 and Table2 should be harmonized: "MSE of Five Different Noise Reduction Algorithms for Anchor Model" and "SNR (dB) of Five Different Noise Reduction Algorithms for Anchor Model" with capitalization problems.
Line 181: The normal vector of the point pi should be ni, not ni and nj; and a similar explanation needs to be added for the point pj .
In addition to the problems noted above, authors should carefully review the entire article to avoid similar problems.
Comments on the Quality of English LanguageOverall, the quality of the English language in this article need minor revision.
Reviewer 4 Report
Comments and Suggestions for Authors
The paper introduces a feature map learning algorithm for point cloud denoising that integrates point-normal constraints, Dirichlet energy, and coupled orthogonality bias terms.
Although the method itself looks reasonable, the assumption of the proposed method needs to be represented. For example, the authors did not explain why they selected these scanning objects in the evaluation. The density of the point clouds from these objects looks high, and it is unclear whether the proposed method can work with low-density point clouds. Thus, the limitations of the proposed method are not described enough
Comments on the Quality of English LanguageN/A
